# Topological Semantic Graph Memory
# for Image-Goal Navigation

**Nuri Kim, Obin Kwon, Hwiyeon Yoo, Yunho Choi, Jeongho Park, and Songhwai Oh**
Department of Electrical and Computer Engineering, ASRI, Seoul National University
{firstname.secondname}@rllab.snu.ac.kr, songhwai@snu.ac.kr

**Abstract:** A novel framework is proposed to incrementally collect landmark-based graph memory and use the collected memory for image goal navigation. Given a target image to search, an embodied robot utilizes semantic memory to find the target in an unknown environment. In this paper, we present a topological semantic graph memory (TSGM), which consists of (1) a graph builder that takes the observed RGB-D image to construct a topological semantic graph, (2) a cross graph mixer module that takes the collected nodes to get contextual information, and (3) a memory decoder that takes the contextual memory as an input to find an action to the target. On the task of an image goal navigation, TSGM significantly outperforms competitive baselines by +5.0-9.0% on the success rate and +7.0-23.5% on SPL, which means that the TSGM finds efficient paths. Additionally, we demonstrate our method on a mobile robot in real-world image goal scenarios. Code is available at https://github.com/rllab-snu/TopologicalSemanticGraphMemory.

**Keywords:** Landmark-Based Navigation, Incremental Topological Memory

## 1  Introduction

Navigation with rich visual observations has been a critical issue in a variety of embodied agent tasks, such as exploration, image goal navigation, and object goal navigation [1–16]. A crucial ingredient for successful visual navigation is to construct a *memory*, which can represent the structure of the environment along with compact visual features for representing high-dimensional visual inputs. A metric-map memory [5, 17] created with SLAM, and a graph memory [8, 9, 18–20] with nodes and edges are the two standard memory construction approaches for navigation algorithms. Even though navigation systems that use metric maps produce powerful results with exact localization and mapping, it is not practical because the navigation agent is susceptible to sensory noises. The topological map, which represents geometric properties and spatial relations of places in the form of a graph, is proposed to construct a map without accurate mapping. Previous visual navigation methods [9, 18] with topological map exploit image features as nodes and edges connecting the nodes in proximity. Since a node indicates a location, the robot's position can be estimated by the nodes in the topological map. Therefore, the graph can be used to navigate successfully even in a real noisy environment.

Unfortunately, using semantic information in topological graph memory presents several difficulties. The first challenge is to incorporate landmarks into a topological graph. As demonstrated by studies [21–23] that show animals navigate utilizing contextual cues from landmarks, the problem of incorporating landmarks, such as object compositions, is a critical issue in visual navigation. A recent research [15] addresses the problem by making an object graph, which connects objects in a field of view of a directional camera to leverage object features. Although it guides an agent to effective action to discover the target utilizing object relationships, it often misses 3D object context information since it only connects objects observed from the same viewpoint. Imagine a goal is given to find a cooking pot, and you know that the cooking pot is usually kept adjacent to an oven. If the oven is nearby but not visible, the agent may miss this crucial information and pick an inefficient path. The more difficult problem is inferring contextual information from objects using geometrically arranged landmarks. There are various advantages to obtaining contextual features

6th Conference on Robot Learning (CoRL 2022), Auckland, New Zealand.

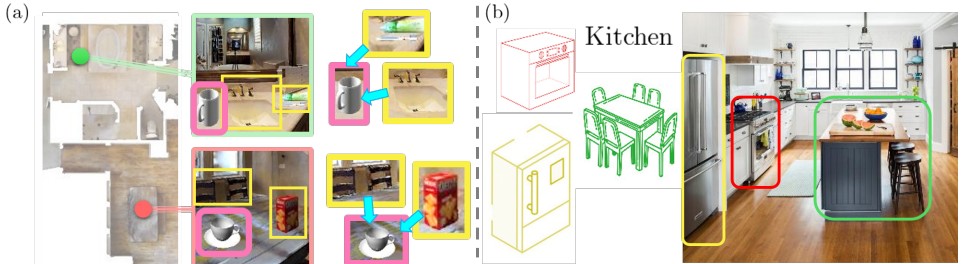

Figure 1: **Importance of semantic contextual information.** (a) When two similar items, e.g., cups, are in different locations, they can be identified individually. (b) A kitchen can be recognized when refrigerator, oven, and dining table are detected.()

from topological graphs. By defining an object through neighboring objects, the contextual representation helps to eliminate the ambiguity of similar but different objects. As shown in Figure 1(a), by describing cups in terms of their neighboring objects, it is possible to distinguish two similar cups. For example, a cup in the kitchen can be perceived as one next to a chair and snack box, while a cup in the bathroom can be shown as one that is near to a toothbrush and washstand. Moreover, a place can be better described through objects. A kitchen, for example, can be defined by the presence of a refrigerator, oven, and dining table (see Figure 1(b)).

To this end, this paper addresses the challenges described above using topological semantic graph memory (TSGM), which has two types of nodes: image nodes and object nodes. Image nodes represent different positions that include image representations, while each object node represents a unique object with its visual representation regardless of viewpoint. If a previously undiscovered image node is identified, it is connected to the previously visited image node, producing a spatially meaningful graph. For each image input, objects are detected using MaskRCNN [24] and then connected to image nodes where the object can be seen. A cross graph mixer, a learnable message-passing network that exchanges object and image information, is then used to make the memory contextual. Using this contextual memory, the agent determines the best strategy for finding the goal. We applied our proposed method to image goal navigation in the Gibson [25] to validate our method. As a result, the proposed method significantly outperforms competitive baselines by +5.0-9.0% on the success rate and +7.0-23.5% on SPL, which means that the TSGM finds efficient paths. On a mobile jackal robot, we demonstrate our method in a real-world environment. The experiment is carried out using episodes where the goal location is not visible and is 5 to 10m away. Despite that the episodes are challenging for image goal navigation, TSGM demonstrates successful navigation.

## 2    Related Work

There has been a lot of research using memory for visual navigation. There are three types of memory formation methods: implicit memory, metric memory, and topological memory.

**Implicit memory.**    Using a recurrent neural network (RNN) as a policy network is a simple method to make an implicit memory [26]. TargetDriven [26] has a vanilla RL policy with a CNN backbone followed by an LSTM which is implicit memory. Since RNN has difficulty backpropagating for a long sequence, RNN is replaced with an explicit memory structure [2, 5, 8, 9, 17, 20].

**Metric memory.**    Active Neural SLAM [17] has a hierarchical structure to explore an environment: global and local policy. The global policy constructs a top-down 2D map and estimates a global goal. It consists of a neural network for flexible output on the input modalities. Given the global goal from the global policy module, a local policy module plans a path to the goal using the simple local navigation algorithm.Exp4nav [5] tackles the exploration problem for navigation. It builds a global metric map, combining egocentric metric maps. Then, to estimate the action, it embeds images with CNN, and a recurrent policy takes the embedding of current observation and the target to output an action.

**Topological memory.**    In [2], LSTM is a policy network that finds the currently located node and derives an action when a topological map is provided. The goal of NeuralPlanner [20] is primarily to

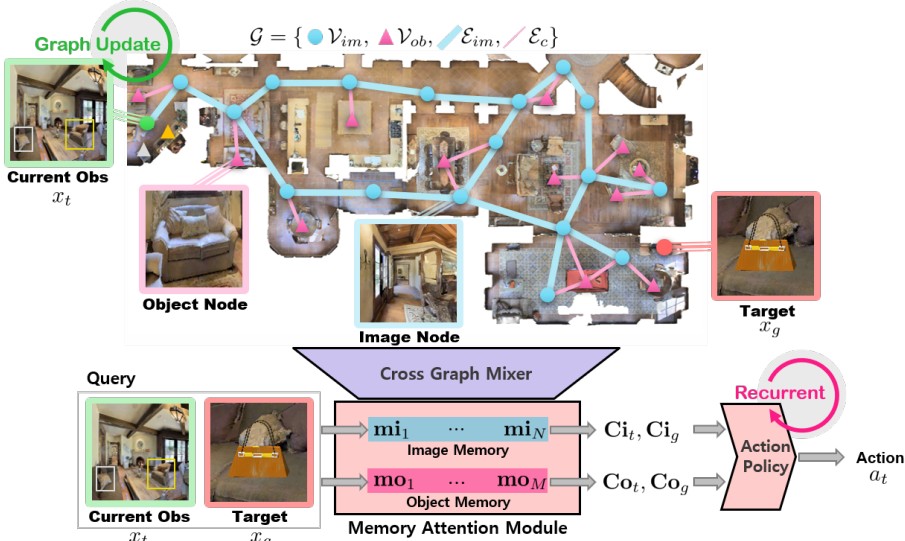

Figure 2: **Overview of the proposed method.** TSGM has a spatially meaningful structure that is generated and updated online. An image node $\mathcal{V}_{im}$ is represented by a blue circle, and an object node $\mathcal{V}_{ob}$ is represented by a pink triangle. The current observation $x_t$ is shown in green, while the goal $x_g$ is highlighted in red. A cross graph mixer module is used to update the constructed graph memory to encode the context of features. Then, the attention module chooses the updated memory by querying the current observation and target node. The action policy network determines the action $a_t$ based on the selected memory. Note that the node position is only used for visualization purposes.

explore the environment and maximize coverage. To this end, a topological map is generated from an exploratory policy during the rollout. Then, a pretrained neural planner calculates the path to the node most similar to the target image. Semi-parametric topological memory (SPTM) [9] forms a topological memory through exploration before training an agent. Then, an agent navigates to the destination based on the topological memory using the Dijkstra algorithm that plans a path to reach the waypoint. Scene memory transformer (SMT) [8] stacks all the visual features of the past observations as a navigation memory. It uses transformer network [6] to process this memory in the context of the current and target observations. In other words, the authors completely replaced the RNN with an attention mechanism, which shows good performance in long-term tasks. Visual graph memory (VGM) [18] constructs a topological visual memory to navigate an environment while not utilizing the landmark information of the scene. No RL no simulator (NRNS) [27] uses models trained using image input without interaction with the simulator. It creates a node for an unexplored area and finds the goal by estimating the geodesic distance to the goal.

Our method builds upon prior topological memory methods to handle the problem of creating and exploiting the object-based semantic graph to find the most efficient path. Once placed in a new environment, our method explores while incrementally constructing a topological graph, inspired by prior work [8, 18, 27]. In contrast to these previous methods, our method employs spatially meaningful landmarks without using exact positions.

## 3 TSGM: Topological Semantic Graph Memory

### 3.1 Problem Statement

We consider the problem of goal-directed exploration for efficient navigation. Given a current observation and a target image to search, an embodied robot aims to find the target (see Figure 2). The current observation contains RGB-D sensory input $x_t$ and detected objects $\{z_1, ..., z_K\} \in \mathbf{z}_t$.

### 3.2 Graph Memory Construction

Let us consider an agent equipped with an RGB-D camera that was dropped into a novel environment that it has never been in before. We want to build and exploit a graph memory that enables

this agent to find the goal in a new environment efficiently. To this end, a semantically structured graph memory, named topological semantic graph memory (TSGM), is built online while an agent traverses the new environment. For example, as shown in Figure 2, when a new node (green circle) is found, the new node is connected to the previous image node (the blue circle connected to the green circle), and the object nodes found at the new node (the green triangles). Since the graph is built with landmarks, recalling the landmarks the agent has seen before will help the agent to discover efficient paths. The TSGM contains two types of nodes and edges, i.e., $\mathcal{G} = \{\mathcal{V}_{im}, \mathcal{V}_{ob}, \mathcal{E}_{im}, \mathcal{E}_c\}$ with image nodes $\mathcal{V}_{im} = \{x_i\}_{i=1}^N$ and its edges $\mathcal{E}_{im}$ with the affinity matrix $A_{im} \in \mathbb{R}^{N \times N}$, object nodes $\mathcal{V}_{ob} = \{z_i\}_{i=1}^M$ where $z_i \in \mathbb{R}^{1 \times D}$, and edges connecting image and object nodes $\mathcal{E}_c$ with the affinity matrix $A_c \in \mathbb{R}^{N \times M}$. $N$ is the number of image nodes, $M$ is the number of object nodes, and $D$ is the dimension of the object representation. The graph expands continuously as the agent explores the environment. The algorithm for constructing TSGM is provided in Supplementary Section 2.

**Object graph construction.** Objects are retrieved using MaskRCNN [24] and added to the graph memory as nodes. When the object is already in memory, the object node is updated with a node with a higher object detection score. If it is not determined that the object is the same, a new object node ($z_k$) is added to the memory, connecting with the corresponding current image node ($x_t$) using the affinity matrix ($A_c$). To connect objects in proximity in the graph memory, we calculate the affinity matrix for object nodes to be connected to objects in the neighbors of the current image node using the image affinity and image-object affinity. Here, since we want a single object node for an object, the trained object encoder using Supervised contrastive learning [28] learns whether the objects are the same object even when the images are shot from various points of view.

**Image graph construction.** To construct an image graph, a pretrained similarity encoder [29] learns image similarity in an unsupervised manner and is then used to determine whether or not the node is already in memory. If an observed node is new, i.e., the node feature is not similar to the existing memory, it is added to the graph memory; otherwise, it is utilized to update the corresponding memory node. While connecting image nodes, a new technique based on objects is utilized, assuming that the likelihood of being at the same place is significantly low if there are few co-visible objects between two image nodes. Therefore, the similarity between the object nodes collected at the current location and the object graph nodes is calculated. We consider two image nodes to represent different locations when there are no similar items between them. If the similarity is low, the observed image node is added to memory; otherwise, the image node is updated to the most recent image representation. When it turns out that the image node is new, it is connected to the previous image node ($x_l$), making the image affinity matrix ($A_{im}$) between $x_l$ and $x_t$ connected.

## 3.3 Cross Graph Mixer

The cross graph mixer is developed from MPNNs [30] to encode scene contexts by combining information from the image and object graphs. The upper row of Figure 3 depicts the image graph, while the lower row represents the object graph. The message passing phase runs for $L$ steps and is defined of message functions $M^l$ and vertex update functions $U^l$, where $l \in \{1, ..., L\}$. The message function $M^l$ is composed of two functions, the self-update function ($S^l$) and the cross-update function ($C^l$). To begin, image and object nodes self-update to obtain contextual representations of nearby locations or objects, respectively:

$$\hat{mi}_v^l = \sum_{w \in \mathcal{N}_i(v)} S_i^l(hi_w^l, A_{im}, g), \quad \hat{mo}_v^l = \sum_{k \in \mathcal{N}_o(v)} S_o^l(ho_k^l, A_{ob}, g), \tag{1}$$

where $\mathcal{N}_i(v)$ and $\mathcal{N}_o(v)$ denote the image/object neighbors of the $v$th node. The node features $h$ are concatenated with the goal feature $g$, and it is embedded into a feature using two-layered neural networks. Then, the self-update message function $S$ makes $\hat{mi}_v^l$ and $\hat{mo}_v^l$ by aggregating connected nodes with the message passing method, which is illustrated in the left column of Figure 3. After self-update, each graph exchanges information with the others to make a complete the messages,

$$mi_v^l = \sum_{k \in \mathcal{N}_o(v)} C_i^l(\hat{mi}_v^l, \hat{mo}_k^l, A_c), \quad mo_v^l = \sum_{w \in \mathcal{N}_i(v)} C_o^l(\hat{mo}_v^l, \hat{mi}_w^l, A_c), \tag{2}$$

where $C_i$ and $C_o$ are cross updating functions that produce $mi_v^l$ and $mo_v^l$, which are cross-updated messages. The cross-update module's central concept is that images are updated through connected

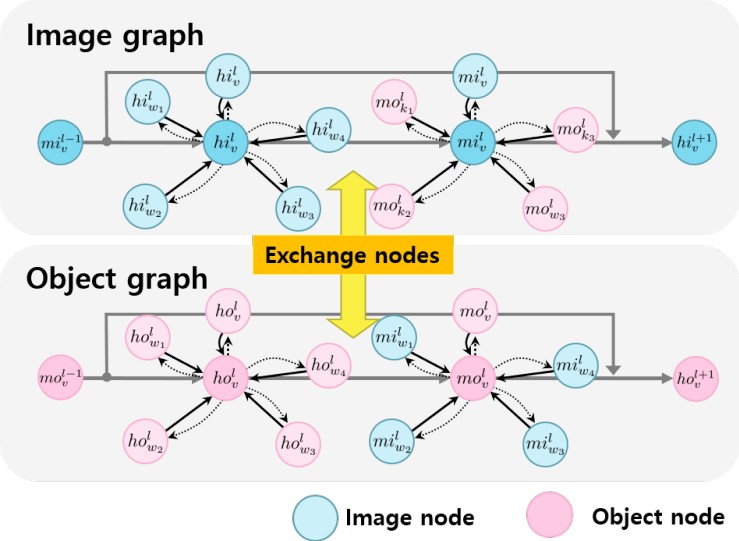

Figure 3: **Cross graph mixer.** The upper row shows the image graph update while the lower row shows the object graph update.

objects, and objects are updated using connected images. To cross-update the graphs, the self-updated image/object features aggregate the connected object/image node features. Then, the update function $U$ transfers messages from object nodes to image nodes and vice versa,

$$hi_v^{l+1} = U_i^l(hi_v^l, mi_v^l), \; ho_v^{l+1} = U_o^l(ho_v^l, mo_v^l). \tag{3}$$

where $U$ is a two-layered neural network that maps information of an object/image to the connected objects/images. The image-object update iterates for $L$ steps to create semantic contextual node features. After $L$ steps of the image-object update iterations, the cross-mixed memory for image graph **mi** and object graph **mo** are generated for each node. The details of the module structure are described in the supplementary material.

### 3.4 Memory Attention Module

TSGM is composed of visited image nodes and object nodes that have been observed. For this, an attention network is used to discover the node closest to the goal among the memory. Since this module utilizes attention, interpolation between visited nodes enables goal features to be extracted from unexplored nodes. Additionally, to find the relative path, the current node is better to be searched. The goal feature $x_g$ is given as a query $q$ when extracting a memory-conditioned goal feature, $\mathbf{Ci}_g, \mathbf{Co}_g$, and the current observation feature $x_t$ is given as an input when selecting the memory-conditioned current feature, $\mathbf{Ci}_t, \mathbf{Co}_t$. Using the decoder module of the transformer network [6], current contextual feature $\mathbf{Ci}_t, \mathbf{Co}_t$ and goal context feature $\mathbf{Ci}_g, \mathbf{Co}_g$ are collected,

$$\mathbf{Ci} = \sigma\Big(\frac{(\mathbf{W}_q q)(\mathbf{W}_k\{\mathbf{mi}_1, ..., \mathbf{mi}_n\}^T)}{\sqrt{d}}\Big)(\mathbf{W}_v q), \mathbf{Co} = \sigma\Big(\frac{(\mathbf{W}_q q)(\mathbf{W}_k\{\mathbf{mo}_1, ..., \mathbf{mo}_n\}^T)}{\sqrt{d}}\Big)(\mathbf{W}_v q), \tag{4}$$

where $\mathbf{W}_v$, $\mathbf{W}_q$, and $\mathbf{W}_k$ are matrix parameters and the set $\{\mathbf{mi}_1, ..., \mathbf{mi}_N\}$ is the image memory, the set $\{\mathbf{mo}_1, ..., \mathbf{mo}_M\}$ is the object memory, and $\sigma$ is a softmax function.

### 3.5 Action Policy Network

Given the contextual feature for goal and current nodes, the action policy network finds an action to reach the goal. For $t$th time step, we encode the sequential information, $\mathbf{h}_a^t = G(\mathbf{h}_c^t, \mathbf{h}_g^t, \mathbf{h}_a^{t-1}, a_{t-1})$, where $G$ is a recurrent neural network and $\mathbf{h}_a^t$ is a $t$th hidden state for action, and $a_{t-1}$ is the previous action of an agent. Then, the action $a_t$ is sampled from the Categorical distribution, i.e. $a_t \sim$ Categorical($\mathbf{W}\mathbf{h}_a^t + \mathbf{b}$), where $\mathbf{W}$ and $\mathbf{b}$ are matrix parameters.

## 3.6 Optimization

**Learning action.** Proximal policy optimization (PPO) [31] is used to learn a policy for picture goal navigation. The policy is trained to maximize the expected return, which is defined as the total reward $\mathbb{E}_\tau[R(s_t, a_t)]$ over time trajectories $\tau = (s_t, a_t)_{t=1}^H$ of the policy's time horizon $H$. When an agent takes action $a_t$ at state $s_t$, $\mathcal{L}_{ppo} = -\mathbb{E}_t\left[\nabla_\theta \log \pi_\theta(a_t|s_t)\hat{A}_t\right]$, where and $\hat{A}_t$ is an estimator of the advantage function at time step $t$, and $pitheta$ is a stochastic policy. We formulated it as a negative log likelihood to the oracle action, $\mathcal{L}_{act} = -\mathbb{E}_{\tau\sim\mathbb{D}}\left[\sum_{t=1}^{T_\tau} a_t^* \log(a_t|x_t)\right]$, where $\tau$ = $(x_t, a_t^*)$ for $t \in \{1, 2, ..., T_\tau\}$, and $a_t^*$ is the oracle action at time step $t$. Since PPO can better estimate return after creating successful episodes, the policy is pretrained with imitation learning and then finetuned with reinforcement learning.

**Auxiliary metrics.** Knowing when to stop is a significant issue since an episode is considered successful if an agent calls the stop action upon seeing the goal. Fortunately, it is well known that auxiliary metrics, such as a progress monitor and goal sensor, can assist an agent in determining when to stop [32]. The progress monitor assesses progress based on the current observation and a target to determine when to press the stop button, which is obtained by calculating the geodesic distance between an agent and the goal. The goal sensor indicates how close the current position is to the goal. If the target is within the success criteria for image goal navigation, the goal sensor produces one. The progress monitor takes the current observation and a target as input and estimates the progress, $\hat{p}_t$. The target sensor takes the same input of the progress monitor and estimates that the current observation containing a target, $\hat{s}_t$. Then, two estimations, $\hat{p}_t$ and $\hat{s}_t$, are optimized with $L_2$ loss, $\mathcal{L}_{aux} = \mathbb{E}_{\tau\sim\mathbb{D}}\left[\sum_{t=1}^{T_\tau} ||s_t^* - \hat{s}_t||_2 + ||p_t^* - \hat{p}_t||_2\right]$, where $s_t^*$ and $p_t^*$ are the ground truths. Since the auxiliary loss is simply added to the action loss, the final loss function for training an agent with imitation learning is $\mathcal{L}_{bc} = \mathcal{L}_{act} + \lambda\mathcal{L}_{aux}$, where $\lambda$ is a balancing parameter. For reinforcement learning, the combined loss $\mathcal{L}_{rl} = \mathcal{L}_{ppo} + \lambda\mathcal{L}_{aux}$, is used.

## 4 Experimental Evaluation

### 4.1 Baselines

We compare our method with baselines with various types of memory.**RGBD + RL** [26] has a vanilla RL policy with a CNN backbone followed by an LSTM adapted from [26]. **Active Neural SLAM** [17] has a metric memory for exploration. To adapt the algorithm to the image goal navigation, we set the output of the global policy to be the relative position to the target when the target is detected using a pretrained target pose estimator. **Exp4nav [5]** tackles the exploration problem.To finetune the image goal navigation task model, we change the coverage reward to the image goal reward. **Neural Planner** [20] is a model adapted the method [20] for image goal navigation from exploration task. **SPTM** [9] creates a topological graph, and then the Dijkstra algorithm then creates a path to a waypoint. **SMT** [8] stacks all the visual features of the past observations and the pose information as a navigation memory. **VGM** [18] builds a topological memory to navigate an environment without using landmark information. **NRNS** [27] trained agent without interaction with the simulator. In order to compare the method fairly, we adapted the method to use a panoramic camera.

### 4.2 Experiment Settings

We evaluate TSGM in the Gibson environment with a habitat simulator, which is photo-realistic. For image observation, we used a panoramic image, the same setting with [18], inspired by the human neuron not recognizing the heading for localizing [33]. The ground truth objects from the Gibson dataset and detected objects from the detector are used for training. For testing, a detector [24] pretrained with a COCO dataset is used. We use a discrete action space defined as $\mathcal{A} = \{$go forward, turn left, turn right, stop$\}$, which is a common choice for navigation problems. The step size of the forward action is $0.25m$, and the rotation angle is set to $10°$ in all experiments. The reward function $R$ is defined using the progress of an agent, using a geodesic distance between an agent and the target, and +10 is given when the agent reaches the goal. Further implementation details of the proposed method are provided in the supplementary material.

Table 1: Comparison of TSGM with memory-based baselines on image goal navigation on Gibson.

| Method | Memory | No Pose | Object | Easy | | Medium | | Hard | | Overall | |
|---|---|---|---|---|---|---|---|---|---|---|---|
| | | | | Success | SPL | Success | SPL | Success | SPL | Success | SPL |
| RGBD + RL [26] | implicit | ✗ | ✗ | 72.5 | 69.5 | 53.1 | 48.6 | 22.3 | 17.7 | 49.3 | 45.3 |
| Active Neural SLAM [17] | metric | ✗ | ✗ | 74.2 | 20.5 | 68.4 | 22.9 | 29.9 | 11.0 | 57.5 | 18.1 |
| Exp4nav [5] | metric | ✗ | ✗ | 70.2 | 61.8 | 60.6 | 52.4 | 46.9 | 38.5 | 59.2 | 50.9 |
| SMT [8] | graph | ✗ | ✗ | 81.9 | 77.4 | 65.6 | 52.2 | 55.6 | 39.7 | 67.7 | 56.4 |
| Neural Planner [20] | graph | ✗ | ✗ | 71.7 | 41.3 | 64.7 | 38.5 | 42.0 | 27.0 | 59.5 | 35.6 |
| SPTM [9] | graph | ✔ | ✗ | 66.5 | 40.6 | 64.2 | 38.5 | 42.1 | 25.4 | 57.6 | 34.8 |
| VGM [18] | graph | ✔ | ✗ | 86.1 | 79.6 | 81.2 | 68.2 | 60.9 | 45.6 | 76.1 | 64.5 |
| TSGM (Ours) | graph | ✔ | ✔ | **91.1** | **83.5** | **82.0** | 68.1 | **70.3** | **50.0** | **81.1** | **67.2** |

Table 2: Comparison of TSGM with image goal navigation baselines on straight/curved episodes on Gibson.

| Path Type | Method | Easy | | Medium | | Hard | | Overall | |
|---|---|---|---|---|---|---|---|---|---|
| | | Success | SPL | Success | SPL | Success | SPL | Success | SPL |
| Straight | NRNS [27] | 67.1 | 57.8 | 52.4 | 41.2 | 32.6 | 22.4 | 50.7 | 40.5 |
| | VGM [18] | 81.0 | 54.4 | 82.0 | 69.9 | 67.3 | 54.4 | 76.7 | 59.6 |
| | TSGM (Ours) | **94.4** | **92.1** | **92.6** | **84.3** | **70.3** | **62.8** | **85.7** | **79.7** |
| Curved | NRNS [27] | 31.7 | 13.0 | 29.0 | 13.6 | 19.2 | 10.4 | 26.6 | 12.3 |
| | VGM [18] | 81.0 | 45.5 | 78.8 | 59.5 | 62.2 | 46.9 | 74.0 | 50.6 |
| | TSGM (Ours) | **93.6** | **91.0** | **89.7** | **77.8** | **64.2** | **55.0** | **82.5** | **74.1** |

**Evaluation metrics.** We use **Success** and **SPL** (success rate over path length) to evaluate the navigation tasks. **Success** is calculated by dividing the total success number by the number of test episodes. We set the success criterion as $1m$ to the goal, a common criterion for image goal navigation. **SPL** multiplies the success rate and the ratio of the shortest path length and traveled path.

**Episode settings.** We divide test episodes into three difficulty levels for image goal navigation: easy, medium, and hard. The difficulty is determined by the geodesic length of an episode. Following [17, 18], we set the length as follows: easy (1.5 – 3m), medium (3 – 5m), and hard (5 – 10m).

### 4.3 Results

**TSGM outperforms baselines.** Table 1 shows the performance of our TSGM and relevant memory-based baselines on test splits of the Gibson dataset. Our TSGM algorithm outperforms the baseline methods in terms of Success and SPL @ $1m$. TSGM improves upon the implicit memory model [26] across splits of Gibson by 31.8% on the success rate. Compared to the VGM [18], our method improved by +5% on the success rate the 4.1% on SPL. To measure performance in various situations, the test episodes are divided according to whether the path is curved or not in Table 2. Our TSGM shows surprising performance improvement in both curved and straight situations. In particular, for an easy episode, the SPL increased quite a surprising amount, 45.5% to 91.0% (100%) for curved episodes and 54.4% to 92.1% (69.3%) for straight episodes compared to the last state-of-the-art result of VGM [18], which can be seen that if an object existing at the target position is present in the field of view, the agent can find the goal position very efficiently. In averaged results in Table 2, TSGM outperforms competitive baselines by 8.5-9.0% on the success rate and 20.2-23.5% improvement on SPL. In summary, TSGM significantly outperforms competitive baselines by 5.0-9.0% on the success rate and 7.0-23.5% improvement on SPL.

**TSGM finds efficient paths.** We checked that the SPL, which indicates the efficiency of the path, is improved a lot. To investigate how the object graph assists the agent in choosing efficient paths, we visualized the paths in Figure 4. When an object graph is not provided, an agent passes the goal without realizing it is a goal, while an agent with an object graph quickly understands that the goal has been achieved and terminates the search (see Figure 4(a)). In Figure 4(b), the agent enters a deadlock when the object graph is not provided. In the case of a TSGM with an object graph, on the other hand, the goal is successfully found with a more efficient path.

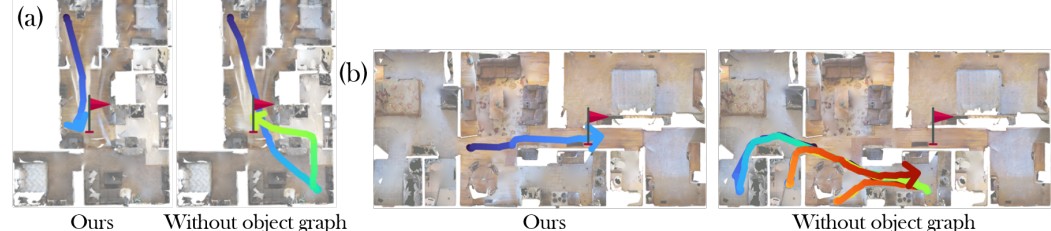

| (a) | | (b) | |
|---|---|---|---|
| Ours | Without object graph | Ours | Without object graph |

Figure 4: **Visualization of path changes when object graph is not given.** Our TSGM is compared with the method that does not have the object graph. The red flag highlights the image goals. The lines with an arrow represent trajectories of the agent with the color changes with time. (a) shows that an inefficient path is created when the object graph is not given. (b) The agent is trapped in the deadlock states without the object graph.

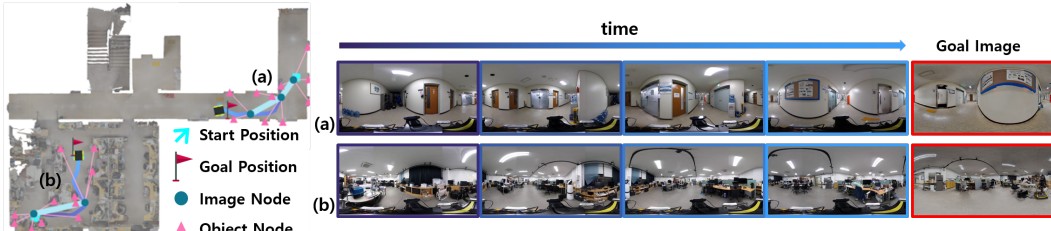

Figure 5: **Visualization of the robot path on real environment.** The color gradation represents the flow of time. The blue arrow symbolizes the starting point, and the red flag indicates the destination. An image node is represented by a blue circle, and an object node is represented by a pink triangle.

**Cross update module is effective.** As shown in Table 3, the ablation study on the graph update was performed, where we only used imitation learning results on hard episodes for the ablation experiments. The object update version indicates the update is done only to the image node to the object node, and the visual update version shows the results when the update is the opposite. Using the semantic graph update shows a 4.5% improvement in success rate and +5.3% on SPL. Surprisingly, the updating object node with visual node shows better results than updating graph with opposite direction, showing +8.0% on the success rate and +6.5%

Table 3: Update rules.

| Update | Success | SPL |
|---|---|---|
| No | 0.533 | 0.393 |
| Visual | 0.578 | 0.446 |
| Object | 0.613 | 0.458 |
| Cross | **0.627** | **0.471** |

on SPL compared to the No update version. Finally, the cross update shows the most powerful results, +9.4% on the success rate and +7.8% on SPL. The results indicate that cross updating improves performance by making image and object features informative.

**TSGM can handle the real noisy world.** Figure 5 shows paths and topological semantic graphs from navigation experiments using a Jackal robot. The episodes examined in the real world are sampled from *hard* episodes with curves. Nonetheless, our agent demonstrated successful navigation by stopping at the correct goal location. The supplementary video shows more paths, with the graph being incrementally built over time.

## 5 Conclusion and Limitation

The proposed method explores while incrementally generating a semantic topological graph using landmarks such as objects. The core idea of our method is derived from animal behavior, which uses landmarks as a navigational cue. Contextual information is taken from the constructed graph to discover an efficient path to the goal. We demonstrate that TSGM performs efficiently and effectively, with significant performance improvements, particularly in SPL. The proposed method is demonstrated using a Jackal mobile robot to show its effectiveness for practical visual navigation in the real-world.

While it is expected to be suitable for finding objects since object contexts are successfully trained in TSGM, a limitation of the proposed method is that no experiments on object-goal navigation are conducted, due to time constraint. We may adapt TSGM to object-goal navigation in the future.

**Acknowledgments**

This work was supported by the Institute of Information & communications Technology Planning & Evaluation(IITP) grant funded by the Korea government(MSIT) (No. 2019-0-01309, Development of AI Technology for Guidance of a Mobile Robot to its Goal with Uncertain Maps in Indoor/Outdoor Environments, (50%) and No. 2019-0-01190, [SW Star Lab] Robot Learning: Efficient, Safe, and Socially-Acceptable Machine Learning, (50%))

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
