# OpenReview forum: "Topological Semantic Graph Memory for Image-Goal Navigation"
_robot-learning.org/CoRL/2022/Conference — CoRL 2022 Oral_

### Official Review · Reviewer_apkv · 2022-07-26

**Originality:** Very Good
**Technical Quality:** Excellent
**Clarity Of Presentation:** Excellent
**Impact:** 4

**Recommendation:**

Strong Accept: I recommend accepting the paper and will argue for my recommendation even if other reviewers hold a different opinion.

**Summary:**

To tackle image goal navigation, this paper proposed a method to represent the memory of landmark images and detected objects in the environment as topological graphs, and to learn navigation action policies using a cross graph mixer and a transformer network. Experiments in a Gibson environment showed that the proposed method significantly outperforms various existing methods.

**Issues:**

- Explanation of what is the advantage/disadvantage to use landmark-based memory against the approach to use time sequence memory.
- Explanation of why the proposed method is not applicable to object goal navigation and the results on e.g., AI2-THOR are not provided.


**Quality Of The Limitations Section:**

Additional details required

**Reviewer Expertise:**

4: The reviewer is confident but not absolutely certain that the evaluation is correct

**Robotics Focus:**

Sufficient demonstration on hardware

**Strengths And Weaknesses:**

Strengths: The paper is very readable and carefully describes the details of the proposed method. The experimental results presented in Table 1, 2, and 3 fully demonstrate the high effectiveness of the proposed method. In addition, the supplemental video shows a real-world experiment using an omnidirectional camera, which is very attractive and well demonstrates the high practicality of the method.

Weaknesses: The idea of using memory and transformer networks in image-goal navigation is not new in itself but rather one of the major recent trends. For example, the following two papers (not cited in this manuscript) take such an approach:
[] Du, Heming, Xin Yu, and Liang Zheng. "VTNet: Visual Transformer Network for Object Goal Navigation." International Conference on Learning Representations. 2020.
[] Rui Fukushima, Kei Ota, Asako Kanezaki, Yoko Sasaki, and Yusuke Yoshiyasu. “Object Memory Transformer for Object Goal Navigation.” IEEE International Conference on Robotics and Automation (ICRA), 2022.
In particular, the latter is similar to the proposed method in that it inputs both object and image memory to the transformer network. The main difference is that they simply store sequential frames in the time axis, whereas the proposed method in this manuscript records landmark information as nodes. The importance of this difference should be discussed in detail (showing experimental results, if possible). Also, as written in the Limitation section on page 8, the evaluation of object goal navigation experiments using such as AI2-THOR is missing. However, the way it is written, it is not clear whether the experiments have not been conducted simply due to lack of time, or whether there is a fundamental theoretical limitation that prevents the proposed method from being applied. This needs to be clarified.


**Summary Of Recommendation:**

There is basically not a single issue that suppresses the significance of the paper, and I see no reason to reject it. The proposed method is interesting and based on a solid theory, and the thorough experimental results and the supplemental video provide very useful insight. For these reasons, I recommend accepting this paper.

### Final Recommendation
The authors have provided reasonable responses in the rebuttal phase, and I was convinced by their responses. The results of the additional experiments with object-goal navigation shown in the general response are also impressive. Therefore, I keep my initial rating, Strong Accept.

---

> ### Author Response · Authors · 2022-08-27
> **Response to Reviewer apkv (Part 2)**
>
>
> ***
> **References**
>
> [1] Du, Heming, Xin Yu, and Liang Zheng. "VTNet: Visual Transformer Network for Object Goal Navigation." International Conference on Learning Representations. 2020.
>
> [2] Rui Fukushima, Kei Ota, Asako Kanezaki, Yoko Sasaki, and Yusuke Yoshiyasu. "Object Memory Transformer for Object Goal Navigation." IEEE International Conference on Robotics and Automation (ICRA), 2022.

---

> ### Author Response · Authors · 2022-08-27
> **Response to Reviewer apkv (Part 1)**
>
>
> We thank reviewer apkv for positive comments and helpful feedback on our work.
> We address some of the reviewer apkv's suggestions in the [general response](https://openreview.net/forum?id=xjTUxBfIzE&noteId=NlgIlIkljvk) to all reviewers above and respond to specific comments below.
>
> **Comparison with time sequence memory methods.**
> > _"Explanation of what is the advantage/disadvantage to use landmark-based memory against the approach to use time sequence memory."_
>
> We explained the advantages/disadvantages of the landmark-based memory below, compared to the time sequence memory.
> * Advantages
>     * Since TSGM only puts a new node into the graph memory by comparing the current observations with the memory, landmark-based memory has less redundancy than time sequence memory.
>     * The time-sequence method combines the image and object features to form a memory in which only a limited number of historical memory can be utilized, while TSGM can use all explicit memory derived from past exploration of the environment.
>     * TSGM has a richer context since adjacent objects can be connected even if they are not observed in the same image. On the other hand, time-sequence memory either assumes that all objects are connected or connecting objects if they are seen in the same image.
>     * When the agent returns to the visited node, TSGM can discover the visited memory node without creating a new node, making the agent detect a loop closure.
>
> * Disadvantages
>     * TSGM needs a pre-trained object discriminator to calculate the similarity between objects.
>
> Additionally, we compared TSGM to two papers [1, 2] that employ the object information mentioned by reviewer apkv.
>
> [1] *VTNet: Visual Transformer Network for Object Goal Navigation.*
> * TSGM has an explicit memory.
>    * In [1], an image and objects form the image are combined to produce a fused representation of a place.
>    * [1] does not have any explicit memory and only has an implicit RNN memory.
>    * TSGM, on the other hand, can explicitly employ previous knowledge gathered while navigating an environment.
> * TSGM can connect out-of-view adjacent objects.
>   * [1] connects object information when detected in the same image.
>   * Since it combines objects detected in the same image, [1] can be seen as a method that only connects objects detected in the same image.
>   * On the other hand, TSGM can connect neighboring object nodes even if the objects are not detected in the same image, resulting in contextually solid representations.
>
> [2] *Object Memory Transformer for Object Goal Navigation.*
> * TSGM has less redundancy in the memory.
>   * In [2], image and object representation are saved in the explicit memory in every time step.
>   * Since TSGM only puts a new node into a graph memory based on the similarity between memory and current observations for both image and object graphs, it has less redundancy than [2].
> * TSGM has a richer context.
>   * [2] utilizes only the preceding $T$ chunks of data.
>   * TSGM, on the other hand, uses all graph memory information derived from past exploration of the environment.
>   * This is achievable due to TSGM's low redundancy.
>
> [1, 2] evaluated their method in AI2Thor, a single-room simulator, while the Gibson environment in which the TSGM is tested is considerably more complicated and vast.
>
> We have included the papers [1, 2] in the supplementary text.
> We could not directly refer these papers in the related work section because of the lack of manuscript space.
> Please see the [general response](https://openreview.net/forum?id=xjTUxBfIzE&noteId=NlgIlIkljvk) about **Q1. Paper organization** where we detail the changes made.
>
>
> **Object goal navigation.**
> >_"Explanation of why the proposed method is not applicable to object goal navigation and the results on e.g., AI2-THOR are not provided."_
>
> As mentioned in the limitation section of the manuscript, our method is expected to be suitable for finding objects since object contexts are successfully trained in TSGM.
> However, due to time constraints, we have only demonstrated that TSGM performs better than the previous methods by focusing on the image goal navigation task.
> We will clarify the reason in the manuscript.
> We have conducted experiments on the object goal navigation task and showed that TSGM could successfully navigate to find objects in Gibson and MP3D environments compared to recent object goal navigation methods.
> The experiments on AI2thor were not available due to a lack of time.
> However, since the Gibson and MP3D are substantially more extensive and more challenging environments than AI2Thor, TSGM is sufficiently validated on the object goal navigation task.
> Please see the [general response](https://openreview.net/forum?id=xjTUxBfIzE&noteId=NlgIlIkljvk) about **Q2. Object goal navigation** for detailed experimental results.

---

### Official Review · Reviewer_T5E1 · 2022-07-28

**Originality:** Very Good
**Technical Quality:** Very Good
**Clarity Of Presentation:** Good
**Impact:** 3

**Recommendation:**

Weak Accept: I recommend accepting the paper, but will not argue for my recommendation if the majority of other reviewers have a different opinion.

**Summary:**

This paper designs a novel topological semantic graph framework to represent the images and objects in the scene, and demonstrates its efficiency on the point goal navigation task in Gibson.  Further, the authors conducts physical experiments and obtain promising results.

**Issues:**

The statement in Section 3 should be reorganized to express the training process more clearly.

**Quality Of The Limitations Section:**

Limitations are addressed clearly

**Reviewer Expertise:**

5: The reviewer is absolutely certain that the evaluation is correct and very familiar with the relevant literature

**Robotics Focus:**

Sufficient demonstration on hardware

**Strengths And Weaknesses:**

Strengths:
1.The novelty of the proposed method which passes information between images and objects and dynamically updates the topological semantic graph in the navigation process.
2.On the point goal navigation task in Gibson, the proposed method outperforms competitive baselines on both SR and SPL.
3.The physical experiments  in real world demonstrates the robustness of the proposed method.

Weaknesses
1.In the comparison experiment, the proposed method is the only one that introduces object feature, which makes it unclear that the performance gain is from the extra information or the topological semantic graph.
2.While the proposed framework constructs a semantic representation of the indoor scene, the authors only test it on the image goal navigation task. More downstream tasks like Object Navigation and Vision Language Navigation could be included.

**Summary Of Recommendation:**

This paper designs a novel topological semantic graph framework to represent the images and objects in the scene. The proposed novel method passes information between images and objects and dynamically updates the topological semantic graph in the navigation process. On the point goal navigation task in Gibson, the proposed method outperforms competitive baselines on both SR and SPL. The significantly improvement of SPL demonstrates that the proposed method could find more efficient path. Further, the  physical experiments  in real world demonstrates the robustness of the proposed method. Some small shortcomings are: In the comparison experiment, the proposed method is the only one that introduces object feature, which makes it unclear that the performance gain is from the extra information or the topological semantic graph; While the proposed framework constructs a semantic representation of the indoor scene, the authors only test it on the image goal navigation task. More downstream tasks like Object Navigation and Vision Language Navigation could be included.

---

> ### Author Response · Authors · 2022-08-27
> **Response to Reviewer T5E1**
>
> We thank reviewer T5E1 for helpful comments and valuable suggestions on our work.
> We address many of reviewer T5E1's suggestions in the [general response](https://openreview.net/forum?id=xjTUxBfIzE&noteId=NlgIlIkljvk) to all reviewers above and respond to specific comments below.
>
> **Analyze the impact of object information.**
> > _"In the comparison experiment, the proposed method is the only one that introduces object feature, which makes it unclear that the performance gain is from the extra information or the topological semantic graph. "_
>
> We have conducted experiments on the impact of objects, where we found that the object helps the localization of an agent.
> Please see the [general response](https://openreview.net/forum?id=xjTUxBfIzE&noteId=NlgIlIkljvk) about **Q3. Analyze the impact of object information** for more details.
>
> **Object goal navigation.**
> > _" While the proposed framework constructs a semantic representation of the indoor scene, the authors only test it on the image goal navigation task. More downstream tasks like Object Navigation and Vision Language Navigation could be included."_
>
> We have conducted experiments on the object goal navigation task and have shown that TSGM can successfully navigate to find objects in Gibson and MP3D environments compared to recent object goal navigation methods.
> Please see the [general response](https://openreview.net/forum?id=xjTUxBfIzE&noteId=NlgIlIkljvk) about **Q2. Object goal navigation** for detailed experimental results.
>
> **Clarify the training process.**
> > _"The statement in Section 3 should be reorganized to express the training process more clearly."_
>
> To summarize the training process,
> 1. The agent builds a graph map, which can be incrementally updated based on the observation.
> 2. Extract latent contextual representation from memory generated from a graph map using the attention method.
> 3. Extract the action by putting the attended memory, current observation, and prior action into the action policy function.
> 4. The action is first optimized with imitation learning with oracle action, and then the agent learns more suited for the environment through reinforcement learning.
>
> We have included the training procedure in Section 1 of the supplementary text to clarify the training process.
> We could not add the training process to Section 3 directly because of the lack of manuscript space.
> Please see the [general response](https://openreview.net/forum?id=xjTUxBfIzE&noteId=NlgIlIkljvk) about **Q1. Paper organization** where we detail the changes made.

---

### Official Review · Reviewer_XjPf · 2022-08-02

**Originality:** Good
**Technical Quality:** Good
**Clarity Of Presentation:** Very Good
**Impact:** 3

**Recommendation:**

Weak Reject: I recommend rejecting the paper, but will not argue for my recommendation if the majority of other reviewers have a different opinion.

**Summary:**

The paper proposes a Topological Semantic Graph Memory (TSGM) for image-goal navigation, which is a graph-memory based method. TSGM outperforms the competitive baselines on success rate and SPL. Physical experiments are also carried to validate the effectiveness on real-world environment.

**Issues:**

Add the analysis how the object features in the object graph help to improve navigation performance.
Add new innovation points.


**Quality Of The Limitations Section:**

Limitations are addressed clearly

**Reviewer Expertise:**

4: The reviewer is confident but not absolutely certain that the evaluation is correct

**Robotics Focus:**

Sufficient demonstration on hardware

**Strengths And Weaknesses:**

Strengths:
1)	This paper proposes a novel topological semantic memory to improve ImageNav performance. The idea of coupling an image graph and object graph is novel and effective. This hierarchical modeling of the agent’s memory is insightful.
2)	Mobile robot demonstration proves the effectiveness.

Weaknesses:
1)	TSGM builds upon other topological memory methods, and it can be considered as small modification of previous algorithms.
2)	It is not convincing that the so called “Cross graph mixer” can actually improve the agent’s performance. It seems that the authors simply add an extra module to VGM. Maybe the extra parameters lead to the improvements, not the “Cross graph mixer”. It would be better to analyze how the object features in the object graph help to improve navigation performance in more detail.


**Summary Of Recommendation:**

The proposed method is proved to be effective and reaches SOTA by adequate experiments on the specific problem of Image-Goal Navigation. However, the method is raised on the basic of prior works and is lacks of innovation. Furthermore, the paper is lack of analysis how the object features in the object graph help to improve navigation performance.

---

> ### Author Response · Authors · 2022-08-27
> **Response to Reviewer XjPf**
>
> We thank reviewer XjPf for constructive and helpful feedback on our work. We address some of reviewer XjPf's suggestions in the [general response](https://openreview.net/forum?id=xjTUxBfIzE&noteId=NlgIlIkljvk) to all reviewers above and respond to specific comments below.
>
> **Technical novelty.**
> > _"TSGM builds upon other topological memory methods, and it can be considered as small modification of previous algorithms."_
>
> Regarding reviewer XjPf's concern about novelty and technical contributions, we emphasize the three main contributions of this paper.
> 1. We provide a novel approach for creating semantic graphs. The object graph connects objects based on a spatial connection, even if the objects are not taken from the same viewpoint. To the best of our knowledge, this is the first attempt to create a topological semantic graph map for the visual navigation task.
> 2. The cross graph mixer enables two graphs to communicate, allowing TSGM to extract contextual representations.
> 3. TSGM performs well in the real world and can explore the scene more efficiently than other topological memory approaches.
>
> Since we are focusing on the image goal navigation task, TSGM was not compared to other navigational algorithms [1,2,3] that use object information as input for object goal navigation.
> However, our method offers significant structural advantages compared to existing navigation systems that employ objects as input.
>
> First, the prior method [2] saves all of the properties of the objects derived from the image in memory without distinguishing if it is a new object, limiting the methods' effectiveness for long-term navigation.
> On the other hand, TSGM only puts a new node into the graph memory by comparing the current observations with the memory.
> Therefore, TSGM has less redundancy on memory than the prior method [2].
>
> Second, past approaches [1,3] combine images and objects from the same perspective and utilize them as a memory.
> As a result, the memory is only connected in time sequences ($A_{im} = 1$, $A_{ob} = 1$) in which all nodes are connected. Also, if object node $i$ is seen in image node $j$, the connection between the image and the object node is connected to the object selected from the same image ($A_c[i,j] = 1$).
> On the other hand, TSGM can connect neighboring object nodes even if the objects are not detected in the same image, resulting in spatially contextual representations.
> Additionally, TSGM is a method which becomes more effective as it moves around the environment since it can update the object representation with the representation in the location where the object is more visible by comparing the detection scores.
>
> **Impact of cross graph mixer.**
> > _"It is not convincing that the so called "Cross graph mixer" can actually improve the agent's performance."_
>
> The cross graph mixer assists object and image nodes in acquiring semantic features.
> In the paragraph *"Cross update module is effective"* of Section 4.3, we conducted an ablation study to show the effectiveness of the cross graph mixer.
> The ablation results (Table 3 of the manuscript) are shown below.
>
>  Method | Success Rate (&#8593;) | SPL (&#8593;)
> ---------------|--------------|--------
>  No Cross Update           | 0.533        | 0.393
>  Visual Update       | 0.578        | 0.446
>  Object Update        | 0.613        | 0.458
>  Cross Update        | **0.627**    | **0.471**
>
> As shown in the table, when cross updates were conducted, the success rate is increased by +9.4%, and the SPL is increased by +7.8% compared to the performance without cross updates.
> Note that there is only a minor parameter difference since only a 2-layer neural network is omitted for stopping for an update (two 2-layer neural networks are not used in the "No Cross Update" setting).
> Therefore, the cross graph mixer assists to improve the agent's performance.
>
> **Analyze the impact of object features on the navigation performance.**
> > _"Add the analysis how the object features in the object graph help to improve navigation performance."_
>
> We have conducted experiments on the impact of objects, where we found that the object helps the localization of an agent.
> Please see the [general response](https://openreview.net/forum?id=xjTUxBfIzE&noteId=NlgIlIkljvk) about **Q3. Analyze the impact of object information** for more details.
> ***
> **References**
>
> [1] Du, Heming, Xin Yu, and Liang Zheng. "VTNet: Visual Transformer Network for Object Goal Navigation." International Conference on Learning Representations. 2020.
> [2] Heming Du, Xin Yu, and Liang Zheng, "Learning Object Relation Graph and Tentative Policy for Visual Navigation" European Conference on Computer Vision (ECCV), 2020.
> [3] Rui Fukushima, Kei Ota, Asako Kanezaki, Yoko Sasaki, and Yusuke Yoshiyasu. "Object Memory Transformer for Object Goal Navigation." IEEE International Conference on Robotics and Automation (ICRA), 2022.

---

### Official Review · Reviewer_dNHx · 2022-08-06

**Originality:** Very Good
**Technical Quality:** Very Good
**Clarity Of Presentation:** Very Good
**Impact:** 3

**Recommendation:**

Weak Accept: I recommend accepting the paper, but will not argue for my recommendation if the majority of other reviewers have a different opinion.

**Summary:**

This paper has proposed a topological semantic graph memory (TSGM) for the image-goal navigation tasks.
The proposed method explores while incrementally generating a semantic topological graph using objects in images. The experimental results showed that the proposed method could improve navigation performance using object and image graphs by cross graph mixer.


**Issues:**

Introduction:
“Since a node indicates a location, the robot’s position is not required.”
I think the author's claims are overstated. The position of the robot may affect the success of the navigation. Topological graphs have several advantages. But aren't its advantages and disadvantages complementary to metric positioning? There are visual navigation approaches that complete navigation only with topological maps. However, some approaches use robot positions.

Fig.1, Importance of objects:
What do you mean? Is there such a thing as confusing cups in different locations? It doesn't seem difficult to determine that they are different cups because they are in different locations. Does this indicate a disadvantage in an approach that uses only topological maps? Are you saying the two cups are not identical because they exist in different locations?
Additionally, it is unclear at this point what exactly you mean by “update”. The object called a “cup” would not be updated. Perhaps the author is trying to say that the topological graph is updated. If that is the case, it would be good to show in Fig. 1 which elements of the topological graph are updated and how.

Some words are provided only as abbreviations. What is the official name? What are these abbreviations? -> ANS, SMT, VGM, NRNS.

spatially meaningful landmarks:
What does it mean to use landmarks without location information? In a metric map such as SLAM, landmarks contain location information. If it is a graph structure without location information, why is the topological graph mapped on an overhead view of the environment in Fig. 2? Perhaps this representation is misleading.


L104: topological semantic graph memory (TSGM):
It appears in the immediately preceding sentence. Abbreviations should be defined at the time of the first appearance.

the image affinity and image-object affinity:
What are these? How are they quantified?

In object and image graph construction, the similarity of features in images is used to determine if objects or nodes are identical. Thereby, if two image nodes do not have similar items, they are determined to represent different locations. But then, whether they are different locations or not should be resolved by using the robot's location information. (As is the case with many VNL approaches.) This architecture has the limitation that it does not use robot location information. This architecture may be disadvantaged because it does not constitute a metric spatial structure. For example, when multiple rooms look almost identical, including the objects present, it seems impossible to discern that they are different rooms.

L190 and 192: Typographical errors?
There appear to be several typographical errors in formulas and variables. The authors would do better to double-check all formulas and variables.

Tables 1 and 2:
Are these results in a seen environment? Or is it an unseen environment?
straight/curved episodes -> What are the differences in these conditions? I could not find where these are described.

What is the number of cross-updates for two different graphs? What is the number of training cycles for PPO, etc.? And what are their computational costs?


**Quality Of The Limitations Section:**

Additional details required

**Reviewer Expertise:**

4: The reviewer is confident but not absolutely certain that the evaluation is correct

**Robotics Focus:**

Highly relevant to robotics but no hardware experiments

**Strengths And Weaknesses:**

Strengths:
- The proposed method can navigate without SLAM or other location estimation or metric maps.
- TSGM has shown performance improvement in pre-experiments for an image-goal navigation task.

Weaknesses:
- The proposed method relies on pre-trained object detectors such as MaskRCNN. As such, it cannot cope with false detections and unknown objects.
- The proposed method is that no experiments on object-goal navigation are conducted.


**Summary Of Recommendation:**

Image-only navigation without the use of location information is a challenging task.
The topological graph construction approach according to image similarity or detected object similarity seems to be effective in a simple way.
This method may also be useful for applications in vision and navigation tasks.

---

> ### Author Response · Authors · 2022-08-27
> **Response to Reviewer dNHx (Part 3)**
>
> **Possible failure case on the graph construction.**
> > _"In object and image graph construction, the similarity of features in images is used to determine if objects or nodes are identical. Thereby, if two image nodes do not have similar items, they are determined to represent different locations. For example, when multiple rooms look almost identical, including the objects present, it seems impossible to discern that they are different rooms."_
>
> As reviewer dNHx said, the graph construction procedure could not distinguish them if multiple rooms seem the same.
> The graph construction can be enhanced by checking memory within a graph hop limit or finding the memory node with neighborhood nodes to distinguish two nodes.
> We appreciate the reviewer's idea to improve the paper's content. We will address the comment in our future work.
>
> **Modification of typographical error.**
> > _" L190 and 192: Typographical errors? There appear to be several typographical errors in formulas and variables."_
>
> The typographical errors are corrected thanks to reviewer dNHx.
> Please see the [general response](https://openreview.net/forum?id=xjTUxBfIzE&noteId=NlgIlIkljvk) about **Q1. Paper organization** where we detail the changes made.
>
> **Clarification of test environments.**
> > _"Tables 1 and 2: Are these results in a seen environment? Or is it an unseen environment?"_
>
> Gibson dataset is divided into 72 train scenes and 14 test scenes for image goal navigation. The results in Tables 1 and 2 are tested in unseen environments.
> We have updated the Supplementary Section 1 to include the test settings.
> Please see the [general response](https://openreview.net/forum?id=xjTUxBfIzE&noteId=NlgIlIkljvk) about **Q1. Paper organization** where we detail the changes made.
>
> **Clarification of episode settings.**
> > _"straight/curved episodes -> What are the differences in these conditions? I could not find where these are described."_
>
> The test episodes used in Table 2 are divided into two types: straight and curved. The ratio of shortest path geodesic-distance to euclidean-distance between the starting and target locations in straight episodes is 1:2, and the rotational difference between the start position and destination is 45°. All other start-goal location pairs are labeled as curved episodes.
> We have updated the sentences in the supplemental text to reflect the differences.
> Please see the [general response](https://openreview.net/forum?id=xjTUxBfIzE&noteId=NlgIlIkljvk) about **Q1. Paper organization** where we detail the changes made.
>
> **The number/size of cross updates and PPO.**
> > _"What is the number of cross-updates for two different graphs? What is the number of training cycles for PPO, etc.? And what are their computational costs?"_
>
> The cross-update is carried out *twice* for both graphs since we wanted to update nodes using updated nodes while utilizing the least computational costs.
> The number of PPO training cycles is set to 10M frames, as can be seen in Supplementary Section 1.
> The computational cost of a cross-update is about $O(N \times M \times D)$, where $N$ indicates the number of image nodes, $M$ represents the number of object nodes, and $D$ represents the dimension of the memory representation.
> The computational complexity is estimated with the cost of matrix multiplication between the cross affinity matrix ($A_c$) and object or image node representations.
>
> We have updated Supplementary Section 1 to include the experimental settings on cross graph mixer.
> Please see the [general response](https://openreview.net/forum?id=xjTUxBfIzE&noteId=NlgIlIkljvk) to all reviewers about **Q1. Paper organization** where we detail the changes made.

---

> ### Author Response · Authors · 2022-08-27
> **Response to Reviewer dNHx (Part 2)**
>
> **Clarify the meaning of Figure 1.**
> > _"Fig.1, Importance of objects: What do you mean? Is there such a thing as confusing cups in different locations? It doesn't seem difficult to determine that they are different cups because they are in different locations. Does this indicate a disadvantage in an approach that uses only topological maps? Are you saying the two cups are not identical because they exist in different locations? Additionally, it is unclear at this point what exactly you mean by "update". The object called a "cup" would not be updated. Perhaps the author is trying to say that the topological graph is updated. If that is the case, it would be good to show in Fig. 1 which elements of the topological graph are updated and how."_
>
> * The meaning of Figure 1.
>   * Figure 1 shows that (a) an object can have a unique representation by utilizing neighboring objects, and (b) a place can be discerned using object nodes.
>   * In Figure 1(a), it is assumed that the two cups are similar and that distinguishing the cup based on the object category and visual attribute is challenging.
>   * Since we designed object nodes in TSGM with the object's visual features and category information, distinguishing the two similar cups without pose may be challenging.
>   * Therefore, an object node can better be connected to the neighboring object nodes to have semantic information.
>   * A self-attention part of the cross graph mixer carries out this procedure.
>
> * The meaning of update.
>     * The **update** refers to the graph convolution of object nodes by connected neighboring object nodes.
>
>
> Figure 1(a) has been modified to show how neighboring objects can help distinguish between two similar-looking objects.
> Also, the description of Figure 1(a) has been modified in response to the reviewer's comments.
> Please see the [general response](https://openreview.net/forum?id=xjTUxBfIzE&noteId=NlgIlIkljvk) about **Q1. Paper organization** where we detail the changes made.
>
> **Abbreviations.**
> > _" Some words are provided only as abbreviations. What is the official name? What are these abbreviations? -> ANS, SMT, VGM, NRNS."_
>
> Here are the full names of the abbreviations used in the paper:
> * SMT: Scene Memory Transformer
> * VGM: Visual Graph Memory
> * NRNS: No RL (Reinforcement Learning) No Simulator
> * ANS: Active Neural SLAM (Simultaneous Localization and Mapping)
>
> SMT, VGM, and NRNS are the official names in their paper, while ANS's official name is Active Neural SLAM.
> We modified *ANS* to *Active Neural SLAM* and made all the full names of the above abbreviations to be explained in the manuscript.
> Please see the [general response](https://openreview.net/forum?id=xjTUxBfIzE&noteId=NlgIlIkljvk) about **Q1. Paper organization** where we detail the changes made.
>
> **Clarification about node position is only used in the Figure for visualization.**
> > _" spatially meaningful landmarks: What does it mean to use landmarks without location information?"_
>
> The actual node locations in Figure 2 are only for visualization purposes; the node location information is not used as an input to TSGM.
> We will clarify that the node position is only used for visualization purposes in the manuscript.
> Please see the [general response](https://openreview.net/forum?id=xjTUxBfIzE&noteId=NlgIlIkljvk) about **Q1. Paper organization** where we detail the changes made.
>
> **Clarification of duplicated expression.**
> > _" L104: topological semantic graph memory (TSGM): It appears in the immediately preceding sentence."_
>
> The sentence has been corrected based on reviewer dNHx's comments.
> Please see the [general response](https://openreview.net/forum?id=xjTUxBfIzE&noteId=NlgIlIkljvk) about **Q1. Paper organization** where we detail the changes made.
>
> **Quantify affinities.**
> > _"the image affinity and image-object affinity: What are these? How are they quantified?"_
>
> The connection between image nodes quantifies image affinity, and image-object affinity is measured by the connected nodes between the image and object graphs, as indicated in Section 3.2 (Graph Memory Construction).
> Specifically, the image affinity matrix $A_{im} \in \mathbb R^{N \times N}$, and the image-object affinity matrix $A_{c} \in \mathbb R^{N \times M}$, where $N$ represents the number of image nodes and the $M$ denotes the number of object nodes. Since the affinity matrices are binary, affinity is one if two nodes are connected and 0 if not.

---

> ### Author Response · Authors · 2022-08-27
> **Response to Reviewer dNHx (Part 1)**
>
> We thank reviewer dNHx for the thorough review and positive feedback on our work.
> We address some of reviewer dNHx's suggestions in the [general response](https://openreview.net/forum?id=xjTUxBfIzE&noteId=NlgIlIkljvk) to all reviewers above and respond to specific comments below.
>
> **Robot experiments.**
> > _"Robotics Focus: Highly relevant to robotics but no hardware experiments"_
>
> The proposed method is demonstrated using a Jackal mobile robot to show its effectiveness for practical visual navigation in the real world.
> The experimental results are described in Figure 5 and the last paragraph of Section 4.
> In addition, the Jackal experiment environment and outcomes can be found in the supplemental video, Supplementary Section 1, and Supplementary Figure 1 to 5.
>
> **Object detector.**
> > _"The proposed method relies on pre-trained object detectors such as MaskRCNN. As such, it cannot cope with false detections and unknown objects."_
>
> The detector used in the paper was trained using the COCO dataset, which includes 80 categories. Thus the majority of the categories are covered.
> Additionally, since the detector was trained on many images (800,000 images), it generalizes effectively.
> For example, if there is an unknown object, the detector will identify the object in a similar category.
>
> Furthermore, TSGM works well even when the detector is not perfect by correcting an object detector's inaccurate detections.
> 1. If an object is not identified, it can be observed from another viewpoint and is connected to the image node adjacent to it.
> 2. When there is a false detection, the weights of edges are trained and utilized in the cross graph mixing module, allowing useless/incorrect objects to be disregarded.
>
> Consequently, TSGM produces reliable results in both simulation and real-world scenarios.
>
> **Object goal navigation.**
> > _"The proposed method is that no experiments on object-goal navigation are conducted."_
>
> We have conducted experiments on the object goal navigation task and have shown that TSGM can successfully navigate to find objects in Gibson and MP3D environments compared to recent object goal navigation methods.
> Please see the [general response](https://openreview.net/forum?id=xjTUxBfIzE&noteId=NlgIlIkljvk) about **Q2. Object goal navigation** for detailed experimental results.
>
> **Modification of the overstated claim.**
> > _"  "Since a node indicates a location, the robot's position is not required." I think the author's claims are overstated. The position of the robot may affect the success of the navigation. Topological graphs have several advantages. But aren't its advantages and disadvantages complementary to metric positioning? There are visual navigation approaches that complete navigation only with topological maps. However, some approaches use robot positions."_
>
> The position of the robot, as mentioned by reviewer dNHx, has a significant impact on the performance of the navigation.
> It is considerably easier to navigate when the robot knows its exact position.
> We corrected the phrase in the manuscript to say the topological graph method could estimate the robot position using the topological map.
> Please see the [general response](https://openreview.net/forum?id=xjTUxBfIzE&noteId=NlgIlIkljvk) about **Q1. Paper organization** where we detail the changes made.

---

### Author Response · Authors · 2022-08-27
**General Response to All Reviewers (Part 3)**


***
**References**

[1] Heming Du, Xin Yu, and Liang Zheng. "VTNet: Visual Transformer Network for Object Goal Navigation." International Conference on Learning Representations. 2020.

[2] Rui Fukushima, Kei Ota, Asako Kanezaki, Yoko Sasaki, and Yusuke Yoshiyasu. "Object Memory Transformer for Object Goal Navigation." IEEE International Conference on Robotics and Automation (ICRA), 2022.

[3] Erik Wijmans, Abhishek Kadian, Ari Morcos, Stefan Lee, Irfan Essa, Devi Parikh, Manolis Savva, and Dhruv Batra, "DD-PPO: Learning Near-perfect PointGoal Navigators From 2.5 Billion Frames." arXiv:1911.00357, 2019.

[4] Brian Yamauchi, "A Frontier-Based Approach for Autonomous Exploration", IEEE International Symposium on Computational Intelligence in Robotics and Automation (CIRA), 1997.

[5] Devendra Singh Chaplot, Dhiraj Gandhi, Saurabh Gupta, Abhinav Gupta, and Ruslan Salakhutdinov. "Learning to Explore using Active Neural SLAM." International Conference on Learning Representations (ICLR), 2020.

[6] Devendra Singh Chaplot, Dhiraj Gandhi, Saurabh Gupta, Abhinav Gupta, and Ruslan Salakhutdinov. "Object Goal Navigation using Goal-Oriented Semantic Exploration." Neural Information Processing Systems (NeurIPS), 2020.

[7] Santhosh Kumar Ramakrishnan, Devendra Singh Chaplot, Ziad Al-Halah, Jitendra Malik, and Kristen Grauman. "PONI: Potential Functions for ObjectGoal Navigation with Interaction-free Learning" IEEE Conference on Computer Vision and Pattern Recognition (CVPR), 2022.

[8] Joel Ye, Dhruv Batra, Abhishek Das, and Erik Wijmans. "Auxiliary Tasks and Exploration Enable ObjectGoal Navigation" IEEE International Conference on Computer Vision (ICCV), 2021.

[9] Oleksandr Maksymets, Vincent Cartillier, Aaron Gokaslan, Erik Wijmans, Wojciech Galuba, Stefan Lee, and Dhruv Batra. "THDA: Treasure Hunt Data Augmentation for Semantic Navigation", IEEE International Conference on Computer Vision (ICCV), 2021.

---

### Author Response · Authors · 2022-08-27
**General Response to All Reviewers (Part 2)**

The results show TSGM is effective for finding objects in Gibson and MP3D environments, even though **TSGM does not use exact pose information**. Note that Gibson and MP3D are much larger and more challenging environments than AI2Thor. Although good results are achieved in our preliminary work for the object goal navigation task, the results are not included in the manuscript since the experiments are not yet fully completed, and the learning process and experimental method could be changed. Therefore, further experimental results on the object goal navigation task will be disclosed in future work. In the [**attached file**](https://openreview.net/pdf?id=NlgIlIkljvk), we explain the metrics used in the object goal navigation experiments and the basic experimental settings.

**Q3. Analyze the impact of object information.**  (Reviewers ecA7, XjPf, and T5E1)
> _"Only the proposed method introduces object features in the comparison experiment, thus it's unclear if the performance boost is from the extra information or the topological semantic graph."_ -ecA7

> _"It would be better to analyze how the object features in the object graph help to improve navigation performance in more detail."_ -XjPf

> _"In the comparison experiment, the proposed method is the only one that introduces object feature, which makes it unclear that the performance gain is from the extra information or the topological semantic graph. "_ -T5E1

In response to comments from reviewers ecA7, XjPf, and T5E1 about showing the impact of object features, we conduct a set of additional experiments.

_**Objects help localization.**_
To demonstrate the influence of object information on performance improvement, we have conducted additional study using attention values extracted from the memory attention module (Section 3.4).
The agent draws the best image memory in the memory attention module, given the current observation.
We computed the success rate of localization using the 1,007 *hard* episodes, also used in Table 1, assuming that localization is successful if the image node selected the memory node closest to the current agent location.
As a result, localization performance is 38.5% without an object node whereas it is 68.2% with an object node, an improvement of 29.7%.


_**Object and image are correlated.**_
To demonstrate that incorporating objects assists in localization, we investigate the correlation between image and object nodes.
We averaged the obtained attention values across the objects in an image since it contains many objects.
Please see Figure 1 of the [attached file](https://openreview.net/pdf?id=NlgIlIkljvk).
In Figure 1 of the [attached file](https://openreview.net/pdf?id=NlgIlIkljvk), the vertical axis represents the average attention score of the object nodes associated with the corresponding ranking's image node.
Since the attention score tends to diminish as the number of image nodes rises, the image node ranking was employed as the horizontal axis by sorting in descending order using the attention value.
Consequently, the image node selected as the current node demonstrated a significant association with a higher object node score.
In summary, we can conclude that the utilization of object nodes makes the agent properly localized, leading an agent to search the path more efficiently while avoiding getting lost.

The [**attached file**](https://openreview.net/pdf?id=NlgIlIkljvk) contains a visualization of attention scores and experimental results demonstrating the relationship between object and image nodes.
We will include details on these new experiments in Supplementary Section 5 of the revised manuscript.

---

### Author Response · Authors · 2022-08-27
**General Response to All Reviewers (Part 1)**

We appreciate the reviewers for their thoughtful and constructive review of our manuscript.
We were encouraged to hear that the reviewers found the method we have presented to be interesting (Reviewer apkv) and that they view our methodology as insightful (Reviewers XjPf and apkv), novel (Reviewers XjPf and T5E1), and effective (Reviewer XjPf).
In response to feedback, we provide a general response here to address points raised by multiple reviewers,
individual responses below to address each reviewer's concerns and an [**attached file**](https://openreview.net/pdf?id=NlgIlIkljvk) about the revised parts in the manuscript.

**Q1. Paper organization.** (Reviewers ecA7, dNHx, XjPf, T5E1, and apkv)

Regarding feedback from reviewers ecA7, dNHx, XjPf, T5E1, and apkv about the organization of our manuscript, we have made the following changes.

* **[dNHx]** Figure 1 (Section 1): We have edited the figure and explanation aboutthe figure to communicate our motivation on the semantic graph structure better.
* **[dNHx]** Figure 2 (Section 1): We have clarified that the node position is only used for visualization and does not used as an input of TSGM.
* **[dNHx]** Optimization (Section 3.6): We have edited typographic errors.
* **[dNHx]** Experimental Settings (Supplementary Section 1): We have added details about the test episodes, evaluation settings, and the details about the cross-update.
* **[apkv]** Object-Based Methods (Supplementary Section 1): We provide additional background on navigation methods utilizing objects.
* **[T5E1]** Training process (Supplementary Section 1): We organized training process more clearly.
* **[ecA7, XjPf, T5E1]** Experiments (Supplementary Section 5): We have significantly expanded our supplemental text to provide analysis and explanations of the impact of the object features.

Note that the Supplementary Section 4, *Path and Graph Visualizations*, is moved to Section 6.
Please see the [**attached file**](https://openreview.net/pdf?id=NlgIlIkljvk) for the detailed changes in the manuscript.

**Q2. Object goal navigation.** (Reviewers ecA7, dNHx, T5E1, and apkv)
> _"No object-goal navigation experiments are proposed."_  -ecA7

> _"The proposed method is that no experiments on object-goal navigation are conducted."_ -dNHx

> _"While the proposed framework constructs a semantic representation of the indoor scene, the authors only test it on the image goal navigation task. More downstream tasks like Object Navigation and Vision Language Navigation could be included."_ -T5E1

> _"As written in the Limitation section on page 8, the evaluation of object goal navigation experiments using such as AI2-THOR is missing."_ -apkv

As mentioned in the limitation section of the manuscript, our method is expected to be suitable for finding objects since object contexts are successfully trained in TSGM.
However, due to time constraints, we have only demonstrated that TSGM performs better than the previous methods by focusing on the image goal navigation task.
We have conducted experiments on the object goal navigation task during the rebuttal period and validated that TSGM outperforms other object goal navigation approaches.
The followings are the experimental results.

* Gibson dataset
    |       Method        |No Pose| Success Rate (&#8593;) |  SPL (&#8593;) |  DTS(&#8595;) |
    |:-------------------|----:|-------------:|-----:|-----:|
    |         BC          |&cross;|         12.2 |  8.3 | 3.90 |
    |        DDPPO [3]       |&cross;|         15.0 | 10.7 | 3.24 |
    |         FBE [4]         |&cross;|         64.3 | 28.3 | 1.78 |
    | Active Neural SLAM [5]  |&cross;|         67.1 | 34.9 | 1.66 |
    |       SemExp [6]        |&cross;|         71.7 | 39.6 | 1.39 |
    |        PONI [7]         |&cross;|         73.6 | **41.0** | **1.25** |
    |        TSGM         |&check;|         **75.1** | 32.7 | 1.48 |

* MP3D dataset
  | Method             |No Pose| Success Rate (&#8593;) |  SPL (&#8593;) |   DTS (&#8595;) |
  |:-------------------|----:|-------------:|-----:|------:|
  | BC                 |&cross;|          3.8 |  2.1 |   7.5 |
  | DDPPO [3]              |&cross;|          8.0 |  1.6 |   6.9 |
  | Red-Rabbit [8]         |&cross;|         34.6 |  1.8 |     - |
  | THDA [9]               |&cross;|         28.4 |  7.9 |   5.6 |
  | FBE [4]                |&cross;|         20.0 |  7.6 |   6.5 |
  | Active Neural SLAM [5] |&cross;|         21.2 |  9.4 |   6.3 |
  | PONI [7]               |&cross;|         27.8 | 12.0 |   5.6 |
  | TSGM               |&check;|         **46.1** | **15.3** |   **4.7** |

---

### Meta-Review · Area_Chair_ecA7 · 2022-08-11

**Recommendation:** Accept (Oral)
**Confidence:** 4

**Metareview:**

Summary
This research proposed a technique to represent landmark images and identified objects as topological graphs and to develop navigation action policies using a cross graph mixer and a transformer network. The suggested method generates an image-based semantic topological graph progressively.  The proposed strategy outperforms existing methods in a Gibson context.

Strengths

1. The proposed technique can traverse without SLAM or other location estimate or metric maps.
 2. This study proposes a topological semantic memory for ImageNav. Combining an image graph and object graph and the  hierarchical memory model is commendable.
3. They provide an effective mobile robot demonstration.
4. The proposed technique reaches SOTA in Image-Goal Navigation trials.
5. The suggested method transmits information between images and objects and dynamically updates the topological semantic graph during navigation.
6.The suggested technique outperforms competitive baselines in SR and SPL for Gibson point goal navigation.
 7. Real-world experiments show the method's robustness.
8.  The paper is well-written and discusses the proposed method. Tables 1, 2, and 3 show the method's effectiveness. The companion movie depicts a real-world experiment utilizing an omnidirectional camera, which is highly appealing and demonstrates the method's feasibility.

Weaknesses
• The proposed approach uses MaskRCNN however false detections and unknown objects can't be handled via this approach.
• No object-goal navigation experiments are proposed.
• It is not clear how many graphs cross-update and how many PPO training cycles and how expensive they are.
What's image affinity and image-object affinity? What's their size?
• TSGM modifies earlier topological memory approaches, however the "Cross graph mixer" doesn't seem to boost agent performance. More insight is needed on how object graph features aid navigation.
• Only the proposed method introduces object features in the comparison experiment, thus it's unclear if the performance boost is from the extra information or the topological semantic graph.
. Although the suggested system creates a semantic indoor scene representation, the authors only test it for picture goal navigation. Object Navigation and Vision Language Navigation could be added.
• Image-goal navigation utilizing memory and transformer networks is a recent technique the reviewers have identified two uncited papers that utilize this approach:
. The evaluation of object goal navigation experiments using AI2-THOR is missing from page 8. The way it's stated, it's unclear whether the experiments haven't been done due to lack of time or a theoretical limitation that prevents the recommended method from being used.

Update: Thank you for the clear explanation to some concerns. The issues raised have been addressed


**Best Paper Nomination:**

No

---

> ### Author Response · Authors · 2022-08-27
> **Response to Area Chair ecA7 (Part 2)**
>
>
> **Impact of cross graph mixer.**
> > _" TSGM modifies earlier topological memory approaches, however the "Cross graph mixer" doesn't seem to boost agent performance."_
>
> The cross graph mixer assists object and image nodes in acquiring semantic features.
> In the paragraph *"Cross update module is effective"* of Section 4.3, we conducted an ablation study to show the effectiveness of the cross graph mixer.
> The ablation results (Table 3 of the manuscript) are shown below.
>
>  Method | Success Rate (&#8593;) | SPL (&#8593;)
> ---------------|--------------|--------
>  No Cross Update           | 0.533        | 0.393
>  Visual Update       | 0.578        | 0.446
>  Object Update        | 0.613        | 0.458
>  Cross Update        | **0.627**    | **0.471**
>
> As shown in the table, when cross updates were conducted, the success rate is increased by +9.4%, and the SPL is increased by +7.8% compared to the performance without cross updates.
> Note that there is only a minor parameter difference since only a 2-layer neural network is omitted for stopping for an update (two 2-layer neural networks are not used in the "no cross update" study).
> Therefore, the cross graph mixer assists to improve the agent's performance.
>
> **Analyze the impact of object information.**
> > _"More insight is needed on how object graph features aid navigation. Only the proposed method introduces object features in the comparison experiment, thus it's unclear if the performance boost is from the extra information or the topological semantic graph."_
>
> We have conducted experiments on the impact of objects, where we found that the object helps the localization of an agent.
> Please see the [general response](https://openreview.net/forum?id=xjTUxBfIzE&noteId=NlgIlIkljvk) about **Q3. Analyze the impact of object information** for more details.
>
> **Comparison with navigation methods that utilize objects.**
> > _"Image-goal navigation utilizing memory and transformer networks is a recent technique the reviewers have identified two uncited papers that utilize this approach."_
>
> We compared TSGM to two papers [1, 2] that employ the object information. Note that the two uncited papers tackles the object-goal navigation task, not image-goal navigation.
>
> [1] *VTNet: Visual Transformer Network for Object Goal Navigation.*
> * TSGM has an explicit memory.
>    * In [1], an image and objects form the image are combined to produce a fused representation of a place.
>    * [1] does not have any explicit memory and only has an implicit RNN memory.
>    * TSGM, on the other hand, can explicitly employ previous knowledge gathered while navigating an environment.
> * TSGM can connect out-of-view adjacent objects.
>   * [1] connects object information when detected in the same image.
>   * Since it combines objects detected in the same image, [1] can be seen as a method that only connects objects detected in the same image.
>   * On the other hand, TSGM can connect neighboring object nodes even if the objects are not detected in the same image, resulting in contextually solid representations.
>
> [2] *Object Memory Transformer for Object Goal Navigation.*
> * TSGM has less redundancy in the memory.
>   * In [2], image and object representation are saved in the explicit memory in every time step.
>   * Since TSGM only puts a new node into a graph memory based on the similarity between memory and current observations for both image and object graphs, it has less redundancy than [2].
> * TSGM has a richer context.
>   * [2] utilizes only the preceding $T$ chunks of data.
>   * TSGM, on the other hand, uses all graph memory information derived from past exploration of the environment.
>   * This is achievable due to TSGM's low redundancy.
>
> We have included the papers [1, 2] in the supplementary text.
> We could not directly refer these papers in the related work section because of the lack of manuscript space.
> Please see the [general response](https://openreview.net/forum?id=xjTUxBfIzE&noteId=NlgIlIkljvk) about **Q1. Paper organization** where we detail the changes made.
>
> ***
> **References**
>
> [1] Du, Heming, Xin Yu, and Liang Zheng. "VTNet: Visual Transformer Network for Object Goal Navigation." International Conference on Learning Representations. 2020.
>
> [2] Rui Fukushima, Kei Ota, Asako Kanezaki, Yoko Sasaki, and Yusuke Yoshiyasu. "Object Memory Transformer for Object Goal Navigation." IEEE International Conference on Robotics and Automation (ICRA), 2022.

---

> ### Author Response · Authors · 2022-08-27
> **Response to Area Chair ecA7 (Part 1)**
>
> We thank Area chair ecA7 for positive comments and helpful feedback on our work.
> We address some of area chair ecA7's suggestions in the [general response](https://openreview.net/forum?id=xjTUxBfIzE&noteId=NlgIlIkljvk) to all reviewers above and respond to specific comments below.
>
> **Object detector.**
> > _"The proposed approach uses MaskRCNN however false detections and unknown objects can't be handled via this approach."_
>
> The detector used in the paper was trained using the COCO dataset, which includes 80 categories. Thus the majority of the categories are covered.
> Additionally, since the detector was trained on many images (800,000 images), it generalizes effectively.
> For example, if there is an unknown object, the detector will identify the object in a similar category.
>
> Furthermore, TSGM works well even when the detector is not perfect by correcting an object detector's inaccurate detections.
> 1. If an object is not identified, it can be observed from another viewpoint and is connected to the image node adjacent to it.
> 2. When there is a false detection, the weights of edges are trained and utilized in the cross graph mixing module, allowing useless/incorrect objects to be disregarded.
>
> Consequently, TSGM produces reliable results in both simulation and real-world scenarios.
>
> **Object goal navigation.**
> > _"No object-goal navigation experiments are proposed.
> > Although the suggested system creates a semantic indoor scene representation, the authors only test it for picture goal navigation.
> > Object Navigation and Vision Language Navigation could be added.
> > The evaluation of object goal navigation experiments using AI2-THOR is missing from page
> > The way it's stated, it's unclear whether the experiments haven't been done due to lack of time or a theoretical limitation that prevents the recommended method from being used."_
>
> We have conducted experiments on the object goal navigation task and have shown that TSGM can successfully navigate to find objects in Gibson and MP3D environments compared to recent object goal navigation methods.
> Please see the [general response](https://openreview.net/forum?id=xjTUxBfIzE&noteId=NlgIlIkljvk) about **Q2. Object goal navigation** for detailed experimental results.
>
> **The number/size of cross updates and PPO.**
> > _"It is not clear how many graphs cross-update and how many PPO training cycles and how expensive they are. What's their size?"_
>
> The cross-update is carried out *twice* for both graphs since we wanted to update nodes using updated nodes while utilizing the least computational costs.
> The number of PPO training cycles is set to 10M frames, as can be seen in Supplementary Section 1.
> The computational cost of a cross-update is about $O(N \times M \times D)$, where $N$ indicates the number of image nodes, $M$ represents the number of object nodes, and $D$ represents the dimension of the memory representation.
> The computational complexity is estimated with the cost of matrix multiplication between the cross affinity matrix ($A_c$) and object or image node representations.
>
> We have updated Supplementary Section 1 to include the experimental settings on cross graph mixer.
> Please see the [general response](https://openreview.net/forum?id=xjTUxBfIzE&noteId=NlgIlIkljvk) to all reviewers about **Q1. Paper organization** where we detail the changes made.
>
> **Quantify affinities.**
> > _"What's image affinity and image-object affinity? What's their size?"_
>
> The affinity matrix quantifies the node connections in the graph, where the value of the matrix is binary.  It has a value of 1 if the nodes are connected and 0 otherwise.
> Specifically, the image affinity matrix is $A_{im} \in \mathbb R^{N \times N}$, and the image-object affinity matrix is $A_{c} \in \mathbb R^{N \times M}$, where $N$ represents the number of image nodes and the $M$ denotes the number of object nodes. Since the affinity matrices are binary, affinity is one if two nodes are connected and 0 if not.
> Please see Section 3.2, Graph Memory Construction, for more details.